# HanDyVQA: A Video QA Benchmark for Fine-Grained Hand-Object Interaction Dynamics

**Masatoshi Tateno**[1,2]    **Gido Kato**[3,2]    **Kensho Hara**[2]
**Hirokatsu Kataoka**[2,4]    **Yoichi Sato**[1]    **Takuma Yagi**[2]

[1]Institute of Industrial Science, The University of Tokyo
[2]National Institute of Advanced Industrial Science and Technology (AIST)
[3]Waseda University [4]Visual Geometry Group, University of Oxford

`{masatate,ysato}@iis.u-tokyo.ac.jp`
`{takuma.yagi,katou.1999,kensho.hara,hirokatsu.kataoka}@aist.go.jp`

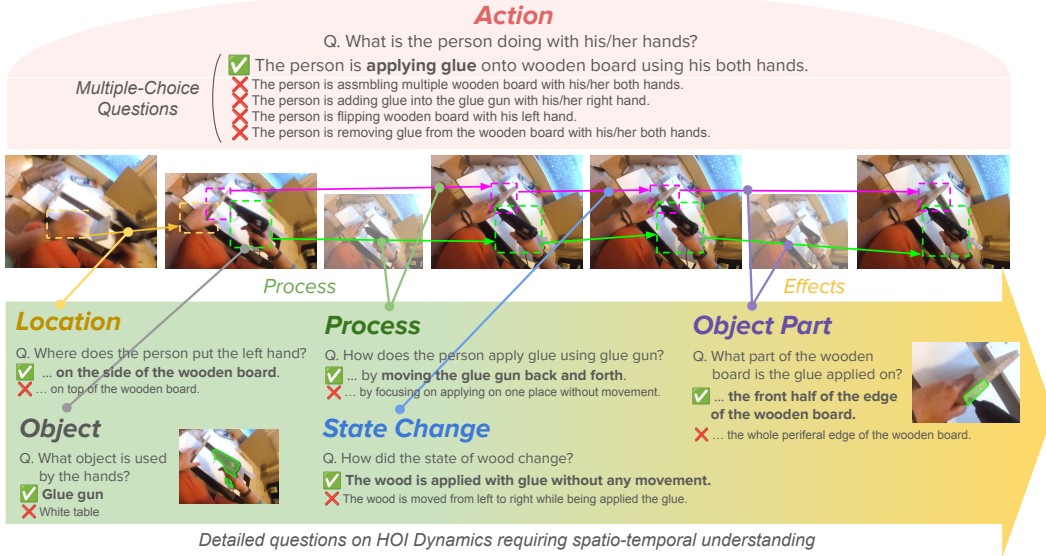

Figure 1: Overview of HandyVQA Dataset.

## Abstract

Hand-Object Interaction (HOI) is inherently a dynamic process, involving nuanced spatial coordination, diverse manipulation styles, and influences on interacting objects. However, existing HOI benchmarks tend to emphasize high-level action recognition and hand/object localization while neglecting the fine-grained aspects of hand-object dynamics. We introduce HanDyVQA, a video question-answering benchmark for understanding the fine-grained spatiotemporal dynamics in hand-object interactions. HanDyVQA consists of six types of questions (Action, Process, Objects, Location, State Change, and Object Parts), totaling 11.7k multiple-choice question-answer pairs and 11k instance segmentations that require discerning fine-grained action contexts, hand-object movements, and state changes caused by manipulation. We evaluated several video foundation models on our benchmark and found that even the powerful Qwen2.5-VL-72B reached only 68.8% average accuracy, uncovering new challenges in component-level geometric and semantic understanding through extensive analyses.

Preprint. Under review.

| | Objective | Source | View | Question Scope | | | | | Answer Type | #Questions | Avg. Duration |
| | | | | HOI | Spatial | Temporal | Process | Effect | | | |
|---|---|---|---|---|---|---|---|---|---|---|---|
| Next-QA [45] | Causal / Temporal / Descriptive | YFCC-100M | TPV | ✗ | ✗ | ✓ | ✗ | ✓ | MC + OP | 52K | 44 s |
| EgoTaskQA [16] | Spatial / Temporal / Causal | LEMMA | FPV | ✗ | ✓ | ✓ | ✗ | ✗ | OP | 40K | 25 s |
| EgoSchema [22] | Long-Term Reasoning | Ego4D | FPV | ✗ | ✗ | ✓ | ✓ | ✗ | MC | 5K | 180 s |
| MVBench [18] | Spatial / Temporal | Mixed | TPV | ✗ | ✓ | ✓ | ✗ | ✗ | MC | 4K | 5–8 s |
| EgoThink [4] | Reasoning / Forecasting / Planning | Ego4D | FPV | ✗ | ✓ | ✓ | ✗ | ✗ | OP | 700 | Single frame |
| HOI-QA [2] | Hand and Object Location Referral | EK/Ego4D | FPV | ✓ | ✓ | ✗ | ✗ | ✗ | OP + BBox | 3.9M | Single frame |
| EgoHOIBench [46] | Action / Objects | Ego4D | FPV | ✓ | ✓ | ✓ | ✗ | ✗ | MC | 30K | 1 s |
| AMB [12] | Long-Term Object Interactions | EK | FPV | ✓ | ✓ | ✓ | ✗ | ✗ | MC | 21K | 20 m |
| HD-EPIC [27] | Fine-grained Video / 3D Understanding | HD-EPIC | FPV | ✓ | ✓ | ✓ | ✓ | ✗ | MC | 26K | Variable |
| HanDyVQA(Ours) | Dynamics / Processes / Effects | Ego4D | FPV | ✓ | ✓ | ✓ | ✓ | ✓ | MC + Seg | 12K | 5 s |

Table 1: Comparison agaist related QA datasets: TPV/FPV refers to third-person-view and first-person-view videos, respectively. MC stands for multiple-choice question-answering and OP represents open-ended question-answering. BBox indicates bounding box, and Seg refers to segmentation.

# 1 Introduction

Hand-Object Interaction (HOI) is inherently a dynamic process [11]. To perform tasks with precision, people choose appropriate tools, carefully coordinate their hands, tools, and objects, and modify the environment to accomplish their goals. Accurately recognizing the spatiotemporal dynamics of hand-object interactions opens up various applications, such as worker assistance [10], dexterous manipulation in robots [36], and motor function analysis [40].

While there has been a surge in hand-object interaction recognition methods and benchmarks in recent years, they tend to focus on either (i) high-level action recognition such as action recognition [7, 17, 13, 46], long-form actions [22], and procedural steps [34, 37, 50] or (ii) low-level localization such as hand-object localization [35, 5, 2] and hand pose estimation [25, 9, 55] while neglecting the semantically rich aspects of hand-object dynamics.

We propose HanDyVQA (**Hand Dy**namics **V**ideo **QA**), a video question-answering benchmark designed to evaluate spatiotemporal reasoning in dynamics of HOI (see Figure 1). HanDyVQA requires an understsnding not only of the actions and objects involved but also of their processes, effects, and component-level changes. We built the benchmark on short video clips extracted from the Ego4D [13] dataset, which fearures diverse and natural hand-object interactions in real-world settings which may not be recorded in intentionally filmed footage. We provide six types of multi-choice question answering (MCQ) tasks totalling 11.7k carefully designed QA pairs that avoid trivial shortcuts, along with referred video object segmentation (RVOS) tasks for two question types (Objects and Object Parts) totalling 11k instances to directly evaluate spatial understanding.

We evaluate existing video-language models to quantify how well they capture various aspects of hand-object interactions. Our results indicate that even the latest foundation models struggle across all categories, achieving only around 61–77% accuracy in MCQ even with the powerful Qwen2.5-VL-72B model. Ablation studies on the number of input frames and image resolution reveals that spatiotemporally dense inputs are necessary to boost the performance. The results on the RVOS task also suggest that current models fail in referring local components finer than object-level.

Furthermore, to advance the understanding of dynamic HOI phenomena, we evaluate HanDyVQA to investigate whether explicitly feeding (i) hand pose, (ii) object tracking, and (iii) object features can enhance the performance or not. The results reveal additional modalities indeed improve performance in many categories, suggesting more sophisticated video encoder design to include local hand-object information towards understanding HOI dynamics.

In summary, our contribution is as follows: (a) We introduce HanDyVQA, a new comprehensive dataset for understanding fine-grained dynamics in HOIs. (b) We conduct an in-depth analysis of how latest video-language models struggle to capture spatiotemporal dynamics and pixel-level reasoning in HOI. (c) We show fine-tuning models with additional hand and object information can enhance the performance, showing the necessity of modeling fine-grained temporal evolution of hands, objects and their components towards further understanding of HOI dynamics.

# 2 Related Work

**Video question answering benchmarks** Conventional benchmarks [47, 52] focus on questions that identify human actions, events, or objects occurring within short video clips of a few seconds. As the field evolves, recent benchmarks have addressed more challenging tasks, such as long-form

video understanding [44, 42, 59, 22]. Some works, such as NExT-QA [45] and TimeLogic QA [39], focus on temporal and causal relationships between multiple actions. MVBench [18] proposes a challenging set of temporal understanding tasks in a multiple-choice QA format that requires watching the entire video by curating major third-person video datasets. HD-EPIC [27] provides a wide variety of fine-grained QAs of egocentric video in a kitchen scenario. However, none of these benchmarks focuses on the fine-grained details of HOIs, including the local coordination between hands and objects, the subtle ways they are handled, and the resulting effects across diverse scenarios.

**Hand-object interaction recognition benchmarks**    Various HOI recognition benchmarks have been proposed for applications such as AR/VR and robotics, with focuses on (i) low-level localization and (ii) high-level actions. For the former, benchmarks have focused on detecting hands and contact objects [35], estimating 3D hand and object poses [14, 3], reconstructing mesh representations [38], and object tracking [1, 12]. AMEGO [12] collects long-term hand and object tracks from the EPIC-KITCHENS dataset and curates a set of questions that require localizing the positions and moments of objects in interaction. For the latter, several benchmarks [22, 46, 2, 4, 12] have been built on egocentric video datasets such as EPIC-KITCHENS [7] and Ego4D [13], since egocentric videos capture close-ups of hands and objects in manipulation. EgoHOIBench [46] introduces an open-vocabulary HOI recognition task that addresses questions about the actions and objects involved in very short (1s) egocentric videos. HOI-QA [2] studies the task of referring to hands and objects in egocentric images, evaluating the relationships between entities and their locations. While these works cover some of the components crucial for HOI understanding, none address the fine-grained nuances of HOIs, such as processes, effects, and component-level spatiotemporal understanding.

**Vision-and-language models for video understanding**    The emergence of dual-encoder vision-language models trained on large-scale image-text pairs [31, 54] has spurred significant advancements in video understanding. Various approaches have been explored, including adapting image-based models for video tasks [21, 49], training models on instructional videos with web-based narrations [24, 23], and pretraining first-person, video-specific models [20, 30, 58]. Following the success of large language models (LLMs), recent video-language models integrate these pretrained visual encoders and LLMs to achieve general video comprehension capabilities across various downstream tasks [2, 57, 6, 51, 41]. Their primary efforts lie in increasing model parameter sizes and expanding training datasets by combining off-the-shelf image and video datasets. However, these multimodal LLM-based models typically employ simple frame-based architectures and rarely account for local entities and spatiotemporal dynamics such as hand poses, manipulated objects, and state or structural changes. HanDyVQA provides a new challenge for developing advanced visual encoders through demanding HOI recognition tasks.

Table 1 shows a comparison against previous datasets. HanDyVQA focuses on the components, processes, and effects of HOIs lasting several seconds, as opposed to instantaneous events (EgoThink, HOI-QA, EgoHOIBench) or long-form events (AMB), and it is the only dataset covering these aspects within the context of HOIs.

# 3    HanDyVQA Benchmark

Our goal is to create a systematic benchmark that evaluates the ability to recognize the spatiotemporal dynamics, processes, and effects present in HOIs. To this end, we define two tasks in our benchmark: (1) Multiple-Choice Question (MCQ) and (2) Referring Video Object Segmentation (RVOS). Given a video and a question, the goal of the MCQ task is to select the correct answer(s) from a set of options, while the RVOS task further requires predicting the segmentation masks corresponding to the correct objects or parts. We define six question categories: Action, Process, Location, Objects, State, and Parts. MCQ samples are provided for all question types, whereas RVOS samples are provided only for Objects and Parts questions.

We opted to adopt the MCQ format over open-ended answers because multiple valid responses can exist for certain types of questions, and MCQ enables quantitative evaluation of fine-grained differences in HOIs by presenting plausible, yet incorrect, alternatives. In this section, we describe details of our data collection process (Section 3.1) and its analysis (Section 3.2).

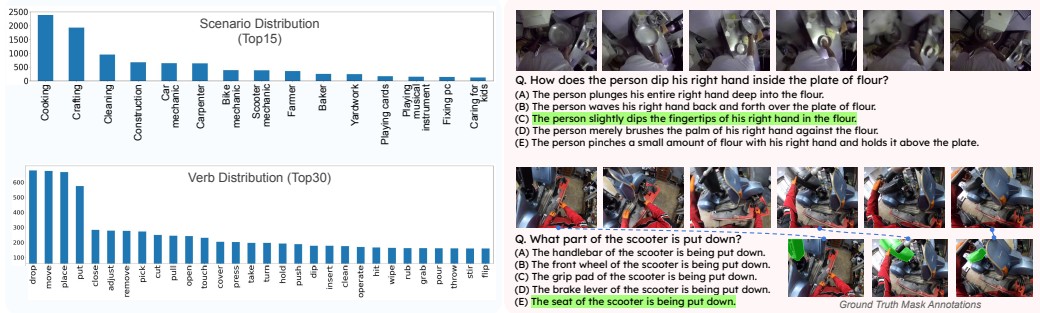

Figure 2: Scenario distribution (left) and example QA pairs of HanDyVQA (right). Sentence with green highlights and green region in images denote correct answer and ground truth masks, respectively.

|  | Action | Process | Location | State | Parts | Objects |
|---|---|---|---|---|---|---|
| #Q | 1978 | 1924 | 1974 | 1940 | 1913 | 1939 |
| #Opt | 5 | 5 | 5 | 5 | 5 | 5.7 |
| #Ans | 1 | 1 | 1 | 1 | 1 | 1.6 |
| #Words | 18.1 | 20.2 | 12.3 | 13.0 | 8.9 | 1.4 |

**(a)** Statistics of QA task. Q: Question, Opt: Options, Ans: Correct answers, Words: Words per option.

|  | #Frames | Avg. Frames per Video | Avg. Centroid Shift (px) | Avg. IoU w/ Adjacent Frames |
|---|---|---|---|---|
| **Objects** | 5546 | 3.36 | 88.28 | 0.08 |
| **Parts** | 5492 | 2.89 | 94.13 | 0.17 |

**(b)** Statistics of segmentation task.

Figure 3: Overview of dataset statistics: (a) Question types and (b) segmentation annotations.

## 3.1 QA collection

We developed a collaborative framework that uses LLMs to propose initial QA candidates, that are carefully refined and verified by humans to ensure quality and diversity.

**Data curation** We build our benchmark on Ego4D [13] as it includes unscripted and realistic hand-object interactions that covers a variety of scenarios and recording locations. To find short video clips capturing moments of HOIs, we utilize the narrations and timestamp information provided in their annotation. Narrations concisely describe the actions performed by the camera wearer, allowing us to automatically determine whether an HOI event is occurring or not within the videos. We feed each narration into LLMs to infer the object in contact and second objects (objects contacted by an in-use tool [5]) for each hand. If we confirm that the camera wearer is manipulating at least one object, we retain the corresponding clip for further use. After curation, we sample 2,000 narrations per category that contain relevant verbs (primary action conducted in the clip) or information to generate questions for each question type. For each narration, we use a 5-second video segment centered around its timestamp, spanning 2.5 seconds before and after the narration. See supplementary for details.

**Question candidate generation** QA pair candidates are automatically generated from narrations using the following templates. **Action:** "What is the person doing with his/her hands?" **Process:** "How does the person [verb] [object]?" **Location:** "Where does the person [verb]" **Objects:** "What object is used by the hands?" **State Change:** "How did the state of [object] change?" **Object Parts:** "What part of [object] is [effect]?"

Verbs and objects are extracted from the narration and inserted into the corresponding placeholders, [verb] and [object]. For **Object Parts** questions, we ask LLMs to infer the plausible objects and effects to be inserted.

**QA refinement by humans** Given the generated questions, annotators verify their validity and revise or reject any that do not match the actual content. Then, they provide a correct answer for each question—listing all plausible objects in the **Objects** category where multiple answers may exist—while ensuring that each answer contains enough detail to be understood without watching the video. Next, wrong answer choice candidates are generated by LLMs based on the question and its

correct answer. Annotators then refine these choices by removing overlaps, improving plausibility, and adding more challenging distractors when necessary. Overall, annotators ensure that all questions, answers, and choices are accurate, sufficiently confusing, and solvable by humans. Examples of challenging questions with confusing choices are shown in Figure 2 right and Figure 5.

**Mask annotation by humans** For the **Objects** and **Object Parts** questions, annotators sampled around three representative frames where the target regions were clearly visible from the video, and annotated the regions corresponding to the correct answer.

### 3.2 Dataset statistics

As a result, 11,668 QA pairs in total are curated for fine-tuning and evaluation. Figure 3 (a) shows the statistics for each question type. **Action** and **Process** exhibit longer descriptions than other categories to explain the nuance of the conducted HOIs. Because often more then one objects are being handled within a 5-seconds clip, an average of 1.6 objects are annotated in **Objects**.

**Diversity in HOI scenarios** As shown in Figure 2, our dataset covers a wide range of video scenarios, including cooking, gardening, cars, and more. We observe a relatively uniform frequency of verbs in the narration annotations, requiring the models to understand various actions and their underlying interactions.

**Distribution of mask annotation** Table 3 (b) shows the number of annotated frames, and the relative movement/spatial overlap between them. Due to the nature of object manipulation and moving cameras in egocentric videos, the segmentation masks shift dynamically over time and space, making them challenging to predict. See supplementary for further analysis.

**Splits** We divided the videos into training, validation, and test sets in a 10 : 5 : 85 ratio, yielding 1.2 K, 0.6 K, and 9.9 K questions, respectively. Only a small portion was set aside as a training/validation set for instruction tuning, allowing the models to learn the required output format while placing greater emphasis on evaluation rather than model training.

## 4 Experiments

| Models | Visual Backbone | Resolution | LLM | Action (Acc) | Process (Acc) | Location (Acc) | State (Acc) | Parts (Acc) | Avg. (Acc) | Objects (AP) |
|---|---|---|---|---|---|---|---|---|---|---|
| Random | – | – | – | 19.3 | 18.8 | 20.5 | 20.0 | 19.4 | 19.6 | 28.5 |
| *Text only models* | | | | | | | | | | |
| GPT-4o (text)[*1] | – | – | GPT-4o | 36.6 | 50.3 | 33.6 | 39.3 | 44.7 | 40.9 | 34.4 |
| *Open-source dual-encoder video-language models* | | | | | | | | | | |
| LaViLa (TSF-L) | TimeSformer | 224x224 | – | 61.2 | 40.0 | 35.8 | 38.5 | 35.6 | 42.2 | 67.0 |
| InternVideo2-Stage2 | Original | 224x224 | – | 40.8 | 30.3 | 29.2 | 34.6 | 30.7 | 33.1 | 36.8 |
| *Open source video-language models w/ integrated LLMs* | | | | | | | | | | |
| VideoLLaMA2.1-7B | SigLip | 384x384 | Qwen2 | 41.1 | 47.1 | 34.4 | 46.3 | 40.0 | 41.8 | 52.1 |
| LLaVa-Video-7B | SigLip | 384x384 | LLaVa-7B | 56.4 | 53.6 | 49.1 | 57.9 | 53.7 | 54.1 | 58.9 |
| mPLUG-Owl3-8B | SigLip | 384x384 | Qwen2 | 56.2 | 51.7 | 44.9 | 54.5 | 47.8 | 51.0 | 59.7 |
| Qwen2.5-VL-7B | Original | 384x384 | Qwen2.5 | 60.2 | 55.0 | 46.9 | 55.5 | 47.4 | 53.0 | 53.0 |
| Qwen2.5-VL-72B | Original | 480x854 | Qwen2.5 | 77.3 | 73.0 | 61.4 | 71.1 | 61.2 | 68.8 | 73.5 |
| *Proprietary vision and language models* | | | | | | | | | | |
| GPT-4o[*1] (vision) | Original | 480x854 | GPT-4o | 60.7 | 64.1 | 50.5 | 58.4 | 57.3 | 58.2 | 62.9 |

Table 2: Comparison of different models across various question types. *1 GPT-4o text/vision refused to answer some questions, providing valid answers to around 87% and 79% of total questions. We report the numbers from valid responses.

| Models | #Frames | Key Features | Action | Process | Location | State | Parts | Avg. | Objects |
|---|---|---|---|---|---|---|---|---|---|
| LaViLa-L | 4 | – | 59.2 | 39.5 | 35.3 | 38.3 | 34.8 | 41.4 | 66.1 |
| HelpingHands-L | 4 | Hand & Object BBox Inference | 56.6 (-2.6) | 36.6 (-2.9) | 34.2 (-1.1) | 39.1 (+0.8) | 34.4 (-0.4) | 40.2 (-1.2) | 67.9 (+1.8) |
| LaViLa-L | 16 | – | 61.2 | 40.0 | 35.8 | 38.5 | 35.6 | 42.2 | 67.0 |
| EgoHOD-L | 16 | Rich Text & Motion Adapter | 59.9 (-1.3) | 37.3 (-2.7) | 37.5 (+1.7) | 41.8 (+3.3) | 35.4 (-0.2) | 42.4 (+0.2) | 74.0 (+7.0) |

Table 3: Comparison of models w/ explicit hand and object modeling

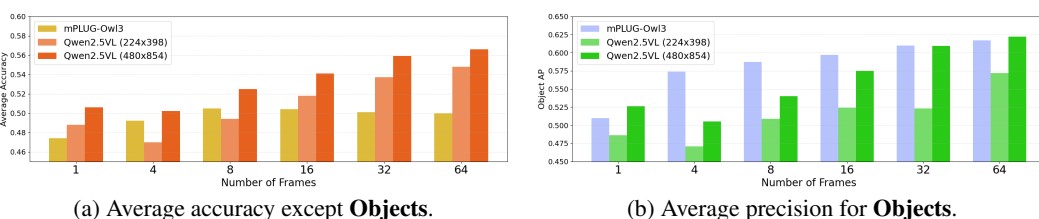

(a) Average accuracy except **Objects**.    (b) Average precision for **Objects**.

Figure 4: Ablations on number of input frames from 1 ($\approx$0.2 fps) to 64 ($\approx$12.8 fps) and resolution.

To reveal the challenges in recognizing the dynamic aspects of HOI in HanDyVQA, we compare the performance of major existing video-language models.

## 4.1 Multi-Choice Questions

We compare the zero-shot performance of representative off-the-shelf video-language models. In addition, we evaluate fine-tuning models with additional modalities of hand and object locations to find directions for future model development.

**Baseline models** We select six open-source video LLMs and one proprietary model, categorized into two types based on their architecture: Dual-Encoder models and LLM-integrated models. **Dual-Encoder models** include LaViLa [58], a video-language model trained on egocentric videos, and InternVideo2-Stage2 [43], whose visual encoder is pre-trained on large-scale video-text pairs. **LLM-integrated models** include VideoLLaMA2.1-7B [6], which specializes in spatio-temporal modeling; LLaVa-Video-7B [19], trained on general and egocentric video datasets; mPLUG-Owl3-8B [51], which efficiently processes long image sequences; and Qwen2.5-VL-7/72B [41], which accepts video inputs with arbitrary resolutions. We also evaluate GPT-4o [15], a proprietary model capable of processing image sequences, in both text-only and vision-enabled settings.

**Implementation details** We uniformly sample 16 frames from each video and use the default input resolution specified for each model. All models are evaluated in a zero-shot setting. Since Qwen2.5-VL supports arbitrary input resolutions, we aligned its input resolution with that of other 7B-scale LLM-integrated models for a fair comparison. For the 72B model, however, we use the full video resolution to showcase its full capability. For dual-encoder models such as LaViLa and InternVideo2, we compute the cosine similarity between the video feature and the text feature of each option, selecting the one(s) with the highest score. For the remaining video LLMs, we provide the video frames along with a prompt listing all options and infer the most probable option(s).

**Evaluation metrics** We report top-1 accuracy for all the categories except **Objects**. We report Average Precision (AP) for **Objects** because it has more than one answers per question.

**Quantitative results** Table 2 shows the quantitative results. Despite preparing answer options unsolvable from text solely, GPT-4o (text) showed moderate results (33–50 pts) than random chance, suggesting some textual bias exists but not enough to solve the task. The dual encoder-based LaViLa trained on the Ego4D dataset outperformed InternVideo2-Stage2, particularly in the **Action** and **Objects** categories, surpassing some LLM-integrated models without text decoder. However, its performance was weaker in other categories, suggesting that LaViLa is specialized to recognize actions and objects. Models with LLM decoders outperformed the text-only baseline, following similar trends observed in general video understanding tasks [33]. Among the 7B-scale models, LLaVA-Video-7B, which is fine-tuned on Ego4D, achieved the highest average accuracy of 54.1%, highlighting the benefits of domain-specific adaptation. Qwen2.5-VL-72B using high resolution images achieved the best overall performance with 68.8% average accuracy, even surpassing GPT-4o (vision) under the same input resolution. However, all models showed limited performance, with top-1 accuracies at most 61% to 77% across categories, suggesting that current large-scale video foundation models still struggle to capture the fine-grained nuances of hand-object interactions.

**Qualitative results** Figure 5 shows examples that most of the models struggled in each category. Major failure cases include (i) missing objects or hand movement mentioned in the question, (ii)

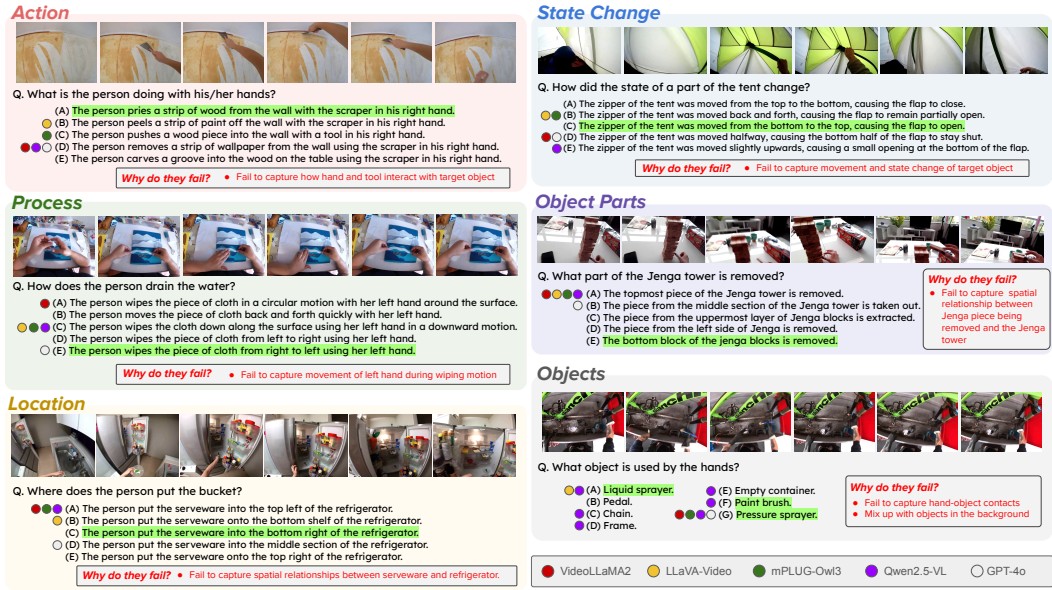

Figure 5: Qualitative results. Sentence with green highlights denote correct answer.

failing to capture the spatiotemporal dynamics between several objects or parts, and (iii) confusing objects spatially close to each other. In summary, the proposed challenging MCQ reveal that large video models exhibit some shortcuts ignoring the fine-grained aspects of HOIs.

**Ablations on number of frames and resolution**  We measured the effect on number of input frames and spatial resolution on mPLUG-Owl3 and Qwen2.5-VL. We changed the number of input frames for mPLUG-Owl3 while keeping the resolution at $384 \times 384$, and varied both the number of input frames and resolutions for Qwen2.5-VL.

As shown in Figure 4, increasing both the number of input frames and resolution was effective. However, for mPLUG-Owl3-8B, performance plateaued beyond 8 frames in all categories except **Objects**, possibly because it has been primarilly trained by 8 frames per video clip. Qwen2.5-VL-7B consistently benefited from incrasing the number of frames across all categories. Increasing input resolution from $224 \times 398$ to $480 \times 854$ led to a consistent improvement in average accuracy of 3.2%–6.8% across all frame settings. The largest gains were seen in the Objects category (6.1%–16.5%), followed by **Action**, **Location**, and **Parts**. The impact was relatively smaller for **Process** and **State** (see Supplemental for full results). These findings suggest that our benchmark requires both spatially and temporally fine-grained details to answer the questions to find the exact moments and locations of the HOI events, compared to the typical settings ($224 \times 224$, 16 frames per clip).

**Evaluation on hand/object-aware models**  Besides the generic video baselines, we also tested HelpingHands [56] and EgoHOD [26], two models expressly designed to capture hand/object-aware features. The former extends the LaViLa visual encoder, while the latter builds on CLIP of a similar size. Both incorporate auxiliary supervision from hand and object bounding box locations—an approach likely well suited to our dataset. The results are shown in Table 3. Both models boosted performance in the **State** and **Objects**, and EgoHOD additionally improving in **Location**. EgoHOD—designed to generate textual descriptions of hand–object motions—outperformed HelpingHands using hand/object location supervision. However, both models reduced accuracy in the remaining categories, suggesting that hand-object location supervision helps with queries about object types and positions but does little for more dynamic aspects.

## 4.2  Referring Video Object Segmentation

**Baseline models**  We compare three baselines: **Sa2VA** [53] **(Frame-wise)**: A multimodal large language model (MLLM) capable of solving both referring image/video segmentation. We input each frame into the Sa2VA model along with the question as a prompt. **Sa2VA (Video)**: We input the

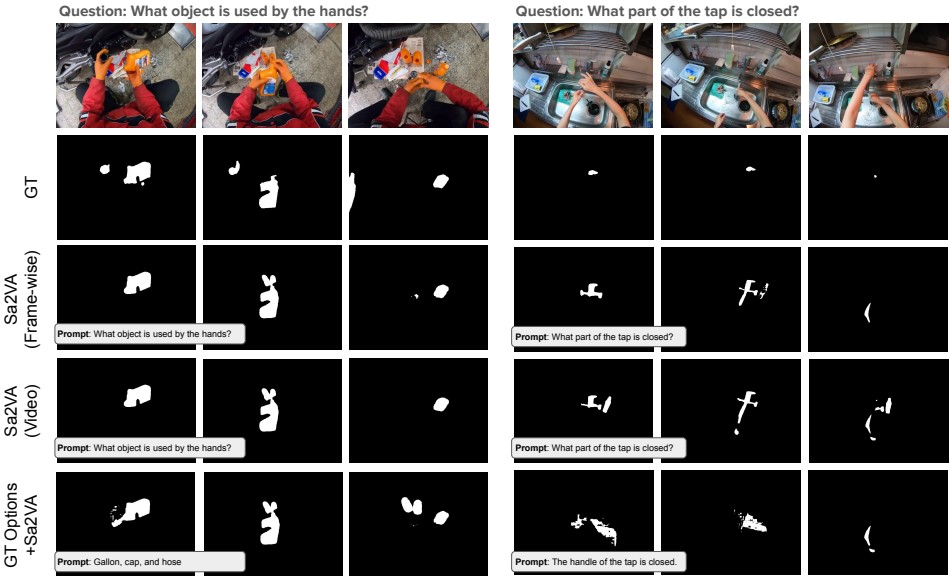

Figure 6: Qualitative results on RVOS. Each black and white image in the bottom shows the ground truth/predicted masks, where white region denotes active regions. The text below each mask image sequence is the textual prompt given to each model.

entire sequence of video frames at once along with the question as a prompt. **Ground Truth Option + Sa2VA**: Use the frame-wise Sa2VA but input the correct answer text instead of questions.

**Evaluation Metrics** Following standard VOS evaluation protocols [28, 48], we use the Jaccard Index ($\mathcal{J}$) and Boundary F-measure ($\mathcal{F}$) computed for each frame and report their average over annotated frames. Furthermore, we categorize the videos into three size-based groups (S/M/L) based on the area of the ground truth masks averaged over each video, and report their average.

**Implementation details** We input 16 frames per video for all the models, ensuring to include the annotated frames while maintaining the frames to be uniformly sampled from the entire video.

**Results** As shown in Table 4, all the models performed significantly worse than those in prior VOS tasks (*e.g.,* 70+ $\mathcal{J}$ in [8]), especially for **Parts** which requires referring to specific parts of objects. We observed different trends in each size group. While giving the GT option led to better scores for larger ground truth masks, single-stage models (Sa2VA frame-wise and video) were better against smaller masks (groups S and M). This is possibly because the ground truth text used in the two-stage approach may be not sufficient to describe the precise region in fine-grained HOIs, often leading to over-segmentation of the target area (*e.g.,* the gallon in Figure 6, left). Sa2VA (frame-wise) achieved slightly higher $\mathcal{J}$ than Sa2VA (video), suggesting that video models struggle to track regions in rapidly moving egocentric videos (*e.g.,* the tap handle in Figure 6, right).

| Models | Objects ($\mathcal{J}$) | | | | Objects ($\mathcal{F}$) | | | | Parts ($\mathcal{J}$) | | | | Parts ($\mathcal{F}$) | | | |
|---|---|---|---|---|---|---|---|---|---|---|---|---|---|---|---|---|
| | S | M | L | All | S | M | L | All | S | M | L | All | S | M | L | All |
| Sa2VA (Frame-wise) | 0.215 | **0.425** | 0.439 | **0.359** | 0.226 | 0.349 | **0.306** | 0.294 | **0.019** | **0.089** | 0.270 | **0.126** | 0.036 | 0.097 | 0.165 | 0.099 |
| Sa2VA (Video) | **0.223** | 0.380 | 0.355 | 0.319 | **0.277** | **0.360** | 0.297 | **0.312** | 0.017 | 0.080 | 0.230 | 0.109 | **0.040** | **0.101** | 0.160 | **0.101** |
| GT option + Sa2VA | 0.076 | 0.239 | **0.464** | 0.259 | 0.088 | 0.185 | 0.268 | 0.182 | 0.011 | 0.060 | **0.284** | 0.119 | 0.024 | 0.070 | **0.172** | 0.089 |

Table 4: Results of RVOS for **Objects** and **Parts** categories. S/M/L denotes groups of videos categorized by the average size of annotated masks within each video.

## 4.3 Integration of HOI cues

We also investigate additional factors to better capture the hand-object dynamics in our dataset. We hypothesize that explicitly feeding spatio-temporally local information about hand manipulations and interacting objects can improve performance compared to relying solely on frame-level features.

| Input | Fine-tune | Action | Process | Location | State | Parts | Avg. | Objects |
|---|---|---|---|---|---|---|---|---|
| Zero-shot (RGB) | No | 40.8 | 30.3 | 29.2 | 34.6 | 30.7 | 33.1 | 36.8 |
| RGB | Yes | 50.0 | 63.6 | 43.4 | 44.8 | 44.7 | 49.3 | 37.1 |
| RGB + BBox | Yes | 48.1 (-3.8%) | 68.2 (+7.2%) | 47.1 (+8.5%) | 47.9 (+6.9%) | 47.5 (+6.3%) | 51.8 (+5.1%) | 37.5 (+1.1%) |
| RGB + Hand | Yes | 49.6 (-0.8%) | 67.1 (+5.5%) | 47.2 (+8.8%) | 49.0 (+9.4%) | 47.0 (+5.2%) | 52.0 (+5.5%) | 37.0 (-0.3%) |
| RGB + Object Feats | Yes | 44.4 (-11.2%) | 68.2 (+7.2%) | 47.1 (+8.5%) | 48.5 (+8.3%) | 45.4 (+1.6%) | 50.7 (+2.8%) | 38.1 (+2.7%) |
| RGB + Hand + BBox | Yes | 50.2 (+0.4%) | 69.1 (+8.6%) | 45.8 (+5.5%) | 48.4 (+8.0%) | 48.2 (+7.8%) | 52.3 (+6.1%) | 37.0 (-0.3%) |
| RGB + Hand + Object Feats | Yes | 42.3 (-15.4%) | 69.5 (+9.3%) | 47.0 (+8.3%) | 46.6 (+4.0%) | 45.8 (+2.5%) | 50.2 (+1.8%) | 38.2 (+3.0%) |
| RGB + Hand + BBox + Object Feats | Yes | 51.5 (+3.0%) | 68.1 (+7.1%) | 46.2 (+6.5%) | 49.4 (+10.3%) | 47.8 (+6.9%) | 52.6 (+6.7%) | 37.8 (+1.9%) |

Table 5: Comparison between different input information. Percentages in Red/Green color indicate performance drop/gain relative to the RGB input.

To test this hypothesis, we chose InternVideo2-Stage2 [43] as our baseline model and fine-tune the model using the training split (1.2 K questions) which could be regarded as a small-scale instruction tuning set. We trained different models by appending addtional branches that input combination of additional cues. Specifically, we considered (i) 3D hand pose information, (ii) bounding box tracklets of manipulated objects, and (iii) their object features.

**Implementation details**   In addition to the Internvideo2 visual/text encoder, separate encoders for each modality consisting of frame-wise MLP and LSTM are introduced. These cues are concatenated with the video feature and passed to a projection layer to match the embedding space. The visual encoder and text encoder of InternVideo2 remain frozen during training, and only the added layers are trained. We input 16 frames per video with a resolution of 224×224. 63-dimensional 3D hand poses are extracted using WiLoR [29]. 4-dimensinal bounding box tracklets of manipulated objects are obtained using AMEGO [12]. 768-dimensional object features are extracted using CLIP [31].

**Results**   Table 5 shows the comparison across different modalities. First, we observe significant improvement by applying fine-tuning (49.3 vs. 33.1 avg. accuracy), especially in the **Process** category that requires answering the detailed process on HOIs. In contrast to the results of EgoHOD and HelpingHands, additional cues boosted performance also in **Process** and **Parts**, suggesting that the standard ViT encoder is suboptimal solving fine-grained tasks in the challenging HanDyVQAdataset.

## 5   Discussion

**What is missing towards understanding HOI dynamics?**   Experimental results show that state-of-the-art models still struggle to capture fine-grained hand–object interactions across categories. Ablation studies and qualitative analyses indicate that these models often miss the precise locations and motions of local components and the interactions between hands and objects, details that are essential for distinguishing key events. Most recent video MLLMs rely on frame-level Vision Transformers that remain frozen during video-text training. However, the improvements observed when fine-tuning the visual encoder with additional modalities suggest considerable room for progress. Modeling below the frame level—by tracking the spatio-temporal evolution of hand movements and object transformations at higher frame rates (*e.g.,* [32]) may further enhance HOI understanding.

**Limitations**   Although we use a multiple-choice format for quantitative evaluation, crafting convincing distractors becomes increasingly difficult as questions grow more specific and detailed. This raises the risk that models exploit subtle textual biases or general commonsense instead of genuine comprehension. A hybrid evaluation that also incorporates free-form answers may be necessary. Moreover, extending the task to predict geometric properties of HOIs (*e.g.,* the positions and shapes of hands, objects and their components) appears to be a promising next step.

## 6   Conclusion

We have proposed HanDyVQA, a new video QA benchmark for evalutating abundant spatiotemporal dynamics in HOIs. Experimental results reveal that strong video-language models struggles in capturing the fine-grained details of HOIs, only achieving at most 61–77% top-1 accuracy in MCQ, and showing poor performance in referring local regions. Ablation studies and modality analysis suggested the need of improvements to model the local spatiotemporal dynamics b/w local components. We hope that HanDyVQA opens up new directions towards future development.

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
