# OpenReview forum: "HanDyVQA: A Video QA Benchmark for Fine-Grained Hand-Object Interaction Dynamics"
_NeurIPS.cc/2025/Datasets_and_Benchmarks_Track — Submitted to NeurIPS 2025 Datasets and Benchmarks Track_

### Official Review · Reviewer_6qKD · 2025-06-24

**Rating:** 4
**Confidence:** 4

**Summary:**

This article proposes a new dataset: HanDyVQA, a video question-answering benchmark for understanding the fine-grained spatiotemporal dynamics in HOI. It defines six question-answering modes (Action, Process, Objects, Location, State Change, and Object Parts) to assist in understanding fine-grained HOI. Experiments show that the current video foundation models are not effective enough under this benchmark, proving the challenge of this task.

**Dataset Code Accessibility:**

Yes

**Ethical Considerations:**

No, there are no or only very minor ethics concerns

**Final Justification:**

Although the author has answered most of my questions, I still think that their method has too much manual design content, which casts doubt on the objectivity of the dataset. However, due to the outstanding contribution of the article, I am willing to raise the score to borderlinie accept.

**Limitations Weaknesses:**

Major:
1. In Sec.3.1, the author asked LLM to infer which part of the object has changed. But in fact, not all objects will undergo local changes. How can LLM generate the corresponding text description for these objects?

2. In Sec.3.1, the author asked LLM to generate alternative wrong answers to the question. However, if LLM generates extremely wrong answers (such as being completely irrelevant to the original question) or the generated answers are very similar to the correct answers, it will lead to the MCQ problem being too easy or difficult. How does the author ensure the consistency of the difficulty of the questions?

3. In Table 4, why is the performance gap with the previous VOS so significant (70+ v.s. <1)? Why not consider the baseline STM in [1]? Is it the performance gap caused by the baseline?

4. In Table 5, why does the performance of location decrease after adding bbox (comparing the third and fifth rows), but parts increases? This is contrary to intuition, because the object bbox should be more conducive to overall localization and not conducive to local reasoning.

Minor:
1. The writing of the article is not clear enough. There is almost no explanation for the most critical part (object effects) in the previous article.

[1] Epic-kitchens visor benchmark: Video segmentations and object relations. 2022.

**Strengths Contributions:**

1. The benchmark in this paper is significantly different from previous datasets: the dataset in this paper focuses on HOI details and gives object effects. Experimental and visualization results show that the current methods have poor reasoning and segmentation results on object effects, which require higher context understanding capabilities of the model. I think this is an interesting and novel challenge.

2. The experimental setting of the article is relatively complete.

---

> ### Author Rebuttal · Authors · 2025-07-31
>
> We appreciate the reviewer’s positive evaluation of the novelty of our benchmark, the completeness of our experimental setup, and for finding the challenge posed by our dataset both interesting and valuable.
>
> We answer the reviewer's questions as follows:
>
> 1. __Concern about generating Object Parts questions using LLMs when objects do not undergo local changes__: We explicitly instruct the LLMs to generate Parts-related questions only when the narrated action in the video is likely to involve changes in a specific part of the object (see prompts in Supplementary Table 9). Moreover, all generated questions are carefully reviewed by human annotators to ensure they are both reasonable and meaningful. By this two-step process, we ensure that the generated VQA questions reliably target the recognition of local object changes.
> 2. __How to ensure the consistency of the difficulty of the questions?__: We ensure that the distractors are challenging enough to require visual understanding and that each question has only one uniquely correct answer. This is achieved through (i) carefully designed prompts and (ii) human verification. As shown in Supplementary Tables 10–15, LLMs are explicitly instructed to generate distractors by slightly altering the correct answer—keeping them close enough to be plausible but incorrect—rather than producing completely irrelevant or overly similar options. Furthermore, human annotators review all generated distractors and revise them if any are potentially correct or too obviously incorrect to be viable choices without watching the video. This process helps maintain a consistent level of difficulty across questions.
> 3. __Reason for performance gap from previous VOS__: There was a formatting inconsistency that may have caused confusion. In the main text, we refer to typical VOS performance using percentage format (e.g., 70+ Jaccard), while in Table 4, our results are reported in decimal format (e.g., 0.359), which corresponds to 35.9%. We will revise the final version to make this distinction clearer and prevent misunderstanding.
>    That said, the gap remains noticeable (70% vs 35%) because our task, RVOS, is fundamentally different from standard VOS, as it performs question answering and segmentation jointly without relying on an initial object mask. This makes the task more challenging and better suited to models capable of video-language grounding and mask prediction, unlike tracking-based models like STM that lack semantic or language grounding.
> 4. __Why Does Adding BBox to (RGB \+ Hand) Decrease Performance for Location but Increase It for Parts?:__ Adding more modalities does not always lead to performance improvements, and can even degrade performance. This is not necessarily because the added features are uninformative, but possibly because the model struggles to learn how to effectively integrate and utilize them, especially when the training data is limited and the modalities exhibit redundancy or conflict. This highlights the need for better multimodal fusion strategies, which we leave as a direction for future work.
>
>    To better understand how much each modality actually contributes to the final performance, we additionally conducted a diagnostic experiment using the model trained with RGB+Hand Pose+BBox+Object Features. At test time, we replaced each modality individually with random values and measured the resulting accuracy drop for each question category. This allows us to assess the model’s reliance on each modality and its relative importance for different types of reasoning.
>
>    As shown in the table below, perturbed Hand Pose leads to the largest performance drop across all question categories when randomized, indicating it is the most informative modality. __Perturbed BBox leads to a performance drop for Action, Location, State, and Objects categories, indicating that spatial information is important for answering these questions. In contrast, it appears to have the opposite impact on Process and Parts, where the object’s spatial position is less informative and can even be noisy due to the inclusion of irrelevant objects.__ Object features have the smallest impact, which is expected since video features already encode rich visual information.
>    We hope this additional analysis provides further insights into the role of each modality and deepens the understanding of how different features contribute to performance.
>
> __Table: Performance Gap When Each Modality Is Perturbed__
> | Perturbed Modality | Action | Process | Location | State | Parts | Objects |
> | :---- | :---- | :---- | :---- | :---- | :---- | :---- |
> | Hand Pose | \-2.5 | \-0.30 | \-0.54 | \-1.45 | \+0.06 | \-1.41 |
> | BBox | \-0.29 | \+0.61 | \-0.24 | \-0.97 | \+0.24 | \-0.60 |
> | Object Feats | \-0.06 | \-0.06 | \+0.24 | \-0.54 | \+0.43 | \-0.04 |
>
> 5. __Explanation for "Effect" mentioned in the paper:__ We will revise to clarify the meaning of the “Effect” annotation, which was underexplained in the previous version despite being a critical component. The “Effect” annotations are directly reflected in our question types: Location, State Change, and Object Parts. This annotation captures the outcomes of hand manipulations on objects—such as changes in position, configuration, or parts—and is conceptually distinct from action recognition, which focuses on the agent’s behavior. These effects are crucial for understanding how object states evolve through interaction, beyond merely identifying the performed action.

---

> > ### Comment · Reviewer_6qKD · 2025-08-06
> >
> > Thank you for the thorough response.
> >
> > The authors have addressed most of my questions. However, I still think that there are too many manual designs in the method. Concerning other reviews' comments, due to the outstanding contribution of this paper, I am willing to raise the score to ba.

---

> > > ### Author Response · Authors · 2025-08-06
> > >
> > > Thank you very much for taking the time to carefully and responsibly review our paper. We sincerely appreciate your thoughtful comments and for reading our rebuttal in detail. We’re grateful that you considered our contributions and are willing to raise your score despite remaining concerns. We will take your feedback into careful consideration in our future work.

---

### Official Review · Reviewer_FxUz · 2025-06-27

**Rating:** 5
**Confidence:** 3

**Summary:**

This paper introduces HanDyVQA, a dataset designed as a benchmark for fine-grained understanding of hand-object interactions. The dataset consists of multiple-choice questions and segmentation tasks, covering various aspects such as HOI understanding, spatial and temporal reasoning, processes, and effects. This provides an evaluation framework for assessing the human-level understanding capabilities of existing models in hand-object interaction scenarios. Interestingly, it reveals that state-of-the-art Video-based LLMs still lacks performance on this benchmark.

**Additional Feedback:**

N/A

**Dataset Code Accessibility:**

Yes

**Dataset Code Comments:**

The dataset is readily available in hugggingface repository.

**Ethical Considerations:**

No, there are no or only very minor ethics concerns

**Final Justification:**

The paper proposes a benchmark, HanDyVQA, for evaluating LLMs (e.g., Qwen2.5, GPT-4o) on their ability to understand fine-grained hand–object interactions. The authors effectively compare recent LLMs and provide valuable insights into their areas for improvement. As LLMs increasingly dominate a wide range of tasks such as image restoration, image generation, and VQA, extending their capabilities to hand–object interaction is an important step. This work offers a valuable opportunity for LLM/MLLM researchers to assess their models and refine them further. The authors have also addressed my concern regarding the lack of reasoning for why LLMs still struggle with fine-grained hand–object interaction. Accordingly, I will maintain my positive rating of Accept.

**Limitations Weaknesses:**

1. According to Table 1, the most discriminative factor of the proposed dataset (HanDyVQA) and other datasets is that “Effect” is included in Question Scope. However, across the paper, what is the “Effect” annotated in the dataset and how it is valuable to understanding hand-object interaction dynamics is not discussed in detail. May I ask why the dataset chose to include “Effect” annotation and how it is annotated.
2. The paper reports detailed results for a vast number of models for different tasks (often regarded as powerful models). However it lacks analysis on why such large-scale models fail. While some experiments and discussion sections suggest that lack of precise localization and reliance on frame-level backbone is a key issue, if that is the problem, then simple fine-tuning would solve the problem. Hope the paper could give more insights on how to improve localization and video-level understanding with respect to what the HandyVQA dataset newly offers to the research community.

**Strengths Contributions:**

1. The proposed dataset provides a challenging benchmark, where even powerful models such as Qwen2.5 exhibit relatively low performance, highlighting substantial room for improvement. This is an interesting finding for the field of hand-object interaction understanding.
2. The paper presents extensive experiments across multiple tasks using large-scale foundation models, LLMs, and task-specific models. These evaluations offer clear insights into the strengths and limitations of current approaches on the proposed benchmark.

---

> ### Author Rebuttal · Authors · 2025-07-31
>
> We sincerely thank the reviewer for highlighting the value of our benchmark in revealing the limitations of current large models in understanding hand-object interactions. We particularly appreciate the reviewer’s remark that our extensive experiments offer clear insights into the strengths and weaknesses of existing approaches.
>
> We address the reviewer’s concerns and questions as follows:
>
> 1. __What does the “Effect” annotation represent in the paper, and why was it included in HanDyVQA?:__ The “Effect” annotation refers to the outcomes of hand manipulations on objects, which is distinct from action recognition. Specifically, it focuses on the resulting changes—such as location shifts, state transitions (e.g., open/closed, intact/broken), and part-level transformations. These are reflected in our question types: Location, State Change, and Object Parts. We included “Effect” questions because recognizing the actual outcome of a manipulation is crucial for applications such as assistive agents or robotics. When an agent aims to perform a specific interaction, it must ensure that the object is transformed into the desired state. We designed HanDyVQA to serve as a fine-grained testbed for evaluating models’ understanding of both hand actions and their resulting object dynamics.
> 2. __Analysis and insights on why such large-scale models fail:__ We explore the underlying patterns behind recognition model failures, based on the additional analyses presented below.
>    1. __Comparing error tendencies between models__ : We measured how often models choose the same wrong option when they fail against Qwen2.5VL. As in the Table below, LaViLa has less overlap (\~30%) except in the Action category. To better understand this, we also conducted a qualitative analysis of LaViLa’s answers. We found that LaViLa—trained primarily on high-level action labels—struggles to generalize to finer-grained queries and often selects entirely irrelevant options.
>    2. __Visual patterns behind model failures (Qwen-2.5-7B baseline)__ : Using the hand-pose and manipulated-object features from §4.3, we computed, for correct vs. incorrect samples per question category, the mean hand-joint displacement, object-bbox displacement, and object-bbox area (Table below). One notable finding is that the difference in hand-joint displacement is clear for the left hand. This may be because the right hand is typically dominant during manipulation, drawing more attention in both training and inference, while the left hand tends to be overlooked:
>          1. __Action__: Recognition is easier when the hand motion is smaller and the object is also relatively small. Very large objects tend to hide the overall action, making it harder to understand.
>          2. __Process__: Smaller movements of both the hand and the object lead to better performance. Large movements make it difficult for the model to follow the process within a limited number of frames.
>          3. __Location__: Larger hand motion helps. It makes the change in position more noticeable and easier for the model to detect.
>          4. __State / Parts__: No clear difference was observed between correct and incorrect cases.
>          5. __Objects__: Smaller hand motion and larger object size are better. Since the goal is to detect the object being manipulated, larger objects are easier to recognize, and less hand movement makes tracking easier.
>
>
> __Table: Answer agreement ratio against Qwen2.5VL (only where both models are wrong)__
> |  | Action | Process | Location | State | Parts |
> | :---- | :---- | :---- | :---- | :---- | :---- |
> | LaViLa | 0.44 | 0.35 | 0.32 | 0.32 | 0.31 |
> | VideoLLaMa2 | 0.57 | 0.47 | 0.67 | 0.60 | 0.63 |
> | LLaVA-Video | 0.53 | 0.54 | 0.61 | 0.58 | 0.53 |
> | mPLUG | 0.59 | 0.57 | 0.67 | 0.55 | 0.56 |
> | GPT\-4o | 0.43 | 0.46 | 0.40 | 0.42 | 0.41 |
>
>
> __Table: Visual patterns of success/failure cases in Qwen2.5VL__
> |  | Left Hand Joint Motion(Sum of all joints) (m/video) | Right Hand Joint Motion(Sum of all joints) (m/video) | BBox Movement (\*e2 px/video) | Bbox Area (e\*4 px^2/video) |
> | :---- | ----- | ----- | ----- | ----- |
> | Action\_Fail | 8.3 | 8.9 | 3.1 | 3.9 |
> | Action\_Suc | 7.7 | 9.1 | 3.0 | 3.4 |
> | Process\_Fail | 8.2 | 8.7 | 3.2 | 3.8 |
> | Process\_Suc | 7.9 | 8.8 | 2.9 | 3.5 |
> | Location\_Fail | 8.2 | 9.6 | 3.1 | 3.7 |
> | Location\_Suc | 8.6 | 9.7 | 3.0 | 3.6 |
> | State\_Fail | 7.9 | 8.6 | 3.0 | 3.6 |
> | State\_Suc | 7.9 | 9.1 | 3.1 | 3.7 |
> | Parts\_Fail | 7.6 | 8.7 | 3.1 | 3.8 |
> | Parts\_Suc | 7.8 | 8.9 | 3.0 | 3.7 |
> | Objects\_Fail | 8.6 | 9.4 | 3.0 | 3.3 |
> | Objects\_Suc | 7.8 | 9.1 | 3.2 | 3.6 |
>
> We also revisit key insights from the main paper to highlight remaining limitations of current vision-language models.
>
> 3. __Video resolution:__ As Figure 4 shows, error rates rise markedly when the input resolution is reduced. This suggests that during both training and inference the models cannot capture the fine-grained spatial cues—e.g. fingertip articulation or small tool edges—required for reliable reasoning.
> 4. __Number of input frames:__ Figure 4 also demonstrates a clear degradation when fewer frames are provided. Notably, while Qwen-2.5 VL shows consistent improvement as more frames are added, mPLUG’s accuracy plateaus beyond eight frames. This suggests that, despite being trained with multiple frames, mPLUG is less capable of exploiting long-range temporal dependencies compared to Qwen-2.5 VL.
> 5. __Training-data domain gap:__ Comparing InternVideo2 (Table 2\) with EgoHOD (Table 3\) reveals that, with an almost identical backbone, the model trained on egocentric footage attains substantially higher accuracy. Including first-person videos in pre-training, therefore, mitigates a critical domain mismatch.

---

> > ### Comment · Reviewer_FxUz · 2025-08-05
> >
> > Thank you for the thorough response.
> >
> > The authors have addressed my questions clearly. I believe this work makes a significant contribution to the field by introducing a new benchmark that enables the evaluation of LLM (which are recent SOTA VQA models) on fine-grained hand-object interaction in videos.
> >
> > I will maintain my rating of Accept.

---

> > > ### Author Response · Authors · 2025-08-05
> > >
> > > Thank you very much for your positive feedback and for recognizing the contribution of our work. We truly appreciate your thoughtful comments and are glad to hear that you found our benchmark valuable.

---

### Official Review · Reviewer_zo7k · 2025-07-03

**Rating:** 4
**Confidence:** 4

**Summary:**

The paper introduces HanDyVQA, a benchmark dataset to evaluate fine-grained understanding of hand-object interaction (HOI) dynamics in egocentric videos. It formulates a multi-choice video question answering (MCQ) task, covering six question types: Action, Process, Object, Object Parts, Location, and State Change. It also includes a referring video object segmentation (RVOS) task to assess spatial grounding of answers. The dataset consists of 11.7k QA pairs and 11k segmentation masks. The paper evaluates a range of state-of-the-art vision-language models, finding that existing models struggle with reasoning over interaction processes and object part localization. The paper also explores the benefit of explicitly integrating HOI-specific cues such as hand pose and object motion information into video encoders.

**Additional Feedback:**

It is better to provide a more detailed analysis of the differences in data collection and annotation procedures among the datasets listed in Table 1.

**Dataset Code Accessibility:**

Yes

**Dataset Code Comments:**

This paper provides complete data collection and annotation code, experimental code, and dataset.

**Ethical Considerations:**

No, there are no or only very minor ethics concerns

**Final Justification:**

The author addresses my concerns in the rebuttal. This work is valuable for fine-grained understanding of hand-object interaction, so my final opinion is borderline accept.

**Limitations Weaknesses:**

- Limited Question Diversity and Interaction Modeling. Compared to multi-level and diverse dense annotations like HD-EPIC, HanDyVQA’s question design is more templated and single-step, lacking deep modeling of fine-grained hand-object interactions and innovation.
- Ambiguity in Task Categories. The task categories (e.g., Action, Process, State Change) have blurred and overlapping boundaries, making it unclear whether the observed model performance differences are due to distinct reasoning challenges or annotation ambiguity.
- Unfair Evaluation. GPT-4o is used for both data generation and evaluation, introducing overlap that may allow the model to exploit its own linguistic patterns during evaluation. As a result, its high performance may come from familiarity of the language style rather than genuine visual reasoning ability.
- Bias from Distractor Options. Although the paper states that the distractor options are carefully designed, they are generated by LLMs. It is concerned that some options may be easily identifiable due to redundancy or stylistic inconsistency, enabling models to select the correct answer without engaging in genuine visual reasoning.
- Lack of Failure Cases Analysis. Although Figure 5 presents failure cases across different question types, the analysis remains superficial, focusing on surface-level descriptions such as “failure to recognize the target action” or “failure to track object movement.” Moreover, the paper does not explore the underlying reasons for the significant performance differences between different models like Qwen2.5 and LaViLa on specific categories.
- Lack of Analysis on HOI cues. The paper does not analyze why some feature combinations (e.g., Hand + BBox + Object) offer only marginal gains, possibly due to redundancy or interference? Additionally, the computational cost of the added branches is not reported.

**Strengths Contributions:**

- The paper is well-written and includes solid experimental results.

---

> ### Author Rebuttal · Authors · 2025-07-31
>
> We appreciate that the reviewer found our paper well-written and that the experimental results are solid.
>
> We answer the reviewer's questions as follows:
>
> 1. __Diversity of Questions and Level of Interaction Modeling__: HanDyVQA focuses on fine-grained and primitive aspects of hand-object interactions, such as detailed hand motions, object state transitions, and part-level changes within a single action. In contrast to HD-EPIC—which emphasizes high-level task understanding in cooking scenarios (e.g., recipes, ingredients, nutrition, and 3D object localization)—our benchmark targets the core physical dynamics of manipulation across diverse domains, not limited to cooking. These two datasets serve complementary goals, with HanDyVQA addressing low-level spatiotemporal reasoning that is often overlooked in existing benchmarks.
>    While HanDyVQA adopts a templated question format for clarity, the answer choices are long and descriptive, requiring models to perform deep HOI reasoning to succeed. For example, Process questions have options averaging over 20 words (Figure 3 (a))—significantly more detailed than those in existing HOI-focused VQA datasets.
> 2. __Ambiguity in Task Categories__: Each question type in HanDyVQA is designed with distinct cognitive demands, primarily realized through the construction of its distractors. For details about the purpose, focus, and distractor design of each category, see Supplementary Tables 16–21. For example, in a video where someone stacks a block, an Action question asks what action is being performed without focusing on the details of how it is done. In contrast, a Process question examines the hand movements used to perform the stacking, and a State Change question asks about how the block’s condition or position has changed as a result. These categories are mutually related, but their distractors are carefully crafted to emphasize different perspectives.
> 3. __Distractor Bias and Evaluation Fairness__: All distractors initially generated by GPT-4o as seeds were refined through human annotation to minimize stylistic bias. During generation, GPT-4o was explicitly instructed to produce distractors with comparable length and granularity to the correct answer, reducing surface-level biases that models could exploit (see prompts in Supplementary Table 10-15). As a result, GPT-4o’s text-only model performed poorly compared to most vision-language models (VLMs) (Table 2), demonstrating that superficial textual biases were successfully mitigated and that visual cues are essential for strong performance. Qwen2.5-VL-72B significantly outperforms GPT-4o (vision), further indicating that model capability, not language style familiarity, drives performance.
> 4. __Failure Cases Analysis:__ We explore the underlying patterns behind recognition model failures, based on the additional analyses presented below.
>    1. __Comparing error tendencies between models__ : We measured how often models choose the same wrong option when they fail against Qwen2.5VL. As in the Table below, LaViLa has less overlap (\~30%) except in the Action category. To better understand this, we also conducted a qualitative analysis of LaViLa’s answers. We found that LaViLa—trained primarily on high-level action labels—struggles to generalize to finer-grained queries and often selects entirely irrelevant options.
>    2. __Visual patterns behind model failures (Qwen-2.5-7B baseline)__ : Using the hand-pose and manipulated-object features from §4.3, we computed, for correct vs. incorrect samples per question category, the mean hand-joint displacement, object-bbox displacement, and object-bbox area (Table below). One notable finding is that the difference in hand-joint displacement is clear for the left hand. This may be because the right hand is typically dominant during manipulation, drawing more attention in both training and inference, while the left hand tends to be overlooked:
>          1. __Action__: Recognition is easier when the hand motion is smaller and the object is also relatively small. Very large objects tend to hide the overall action, making it harder to understand.
>          2. __Process__: Smaller movements of both the hand and the object lead to better performance. Large movements make it difficult for the model to follow the process within a limited number of frames.
>          3. __Location__: Larger hand motion helps. It makes the change in position more noticeable and easier for the model to detect.
>          4. __State / Parts__: No clear difference was observed between correct and incorrect cases.
>          5. __Objects__: Smaller hand motion and larger object size are better. Since the goal is to detect the object being manipulated, larger objects are easier to recognize, and less hand movement makes tracking easier.
>
>
> __Table: Answer agreement ratio against Qwen2.5VL (only where both models are wrong)__
> |  | Action | Process | Location | State | Parts |
> | :---- | :---- | :---- | :---- | :---- | :---- |
> | LaViLa | 0.44 | 0.35 | 0.32 | 0.32 | 0.31 |
> | VideoLLaMa2 | 0.57 | 0.47 | 0.67 | 0.60 | 0.63 |
> | LLaVA-Video | 0.53 | 0.54 | 0.61 | 0.58 | 0.53 |
> | mPLUG | 0.59 | 0.57 | 0.67 | 0.55 | 0.56 |
> | GPT\-4o | 0.43 | 0.46 | 0.40 | 0.42 | 0.41 |
>
>
> __Table: Visual patterns of success/failure cases in Qwen2.5VL__
> |  | Left Hand Joint Motion(Sum of all joints) (m/video) | Right Hand Joint Motion(Sum of all joints) (m/video) | BBox Movement (\*e2 px/video) | Bbox Area (e\*4 px^2/video) |
> | :---- | ----- | ----- | ----- | ----- |
> | Action\_Fail | 8.3 | 8.9 | 3.1 | 3.9 |
> | Action\_Suc | 7.7 | 9.1 | 3.0 | 3.4 |
> | Process\_Fail | 8.2 | 8.7 | 3.2 | 3.8 |
> | Process\_Suc | 7.9 | 8.8 | 2.9 | 3.5 |
> | Location\_Fail | 8.2 | 9.6 | 3.1 | 3.7 |
> | Location\_Suc | 8.6 | 9.7 | 3.0 | 3.6 |
> | State\_Fail | 7.9 | 8.6 | 3.0 | 3.6 |
> | State\_Suc | 7.9 | 9.1 | 3.1 | 3.7 |
> | Parts\_Fail | 7.6 | 8.7 | 3.1 | 3.8 |
> | Parts\_Suc | 7.8 | 8.9 | 3.0 | 3.7 |
> | Objects\_Fail | 8.6 | 9.4 | 3.0 | 3.3 |
> | Objects\_Suc | 7.8 | 9.1 | 3.2 | 3.6 |
>
>
> 5. __Analysis on HOI cues:__ Adding more modalities does not always lead to performance improvements, and can even degrade performance. This is not necessarily because the added features are uninformative, but possibly because the model struggles to learn how to effectively integrate and utilize them, especially when the training data is limited and the modalities exhibit redundancy or conflict. This highlights the need for better multimodal fusion strategies, which we leave as a direction for future work.
>
>    To better understand how much each modality actually contributes to the final performance, we additionally conducted a diagnostic experiment using the model trained with RGB+Hand Pose+BBox+Object Features. At test time, we replaced each modality individually with random values and measured the resulting accuracy drop for each question category. This allows us to assess the model’s reliance on each modality and its relative importance for different types of reasoning.
>
>    As shown in the table below, perturbed Hand Pose leads to the largest performance drop across all question categories when randomized, indicating it is the most informative modality. Perturbed BBox leads to a performance drop for Action, Location, State, and Objects categories, indicating that spatial information is important for answering these questions. In contrast, it appears to have the opposite impact on Process and Parts, where the object’s spatial position is less informative and can even be noisy due to the inclusion of irrelevant objects. Object features have the smallest impact, which is expected since video features already encode rich visual information.
>    We hope this additional analysis provides further insights into the role of each modality and deepens the understanding of how different features contribute to performance.
>
> __Table: Performance Gap When Each Modality Is Perturbed__
> | Perturbed Modality | Action | Process | Location | State | Parts | Objects |
> | :---- | :---- | :---- | :---- | :---- | :---- | :---- |
> | Hand Pose | \-2.5 | \-0.30 | \-0.54 | \-1.45 | \+0.06 | \-1.41 |
> | BBox | \-0.29 | \+0.61 | \-0.24 | \-0.97 | \+0.24 | \-0.60 |
> | Object Feats | \-0.06 | \-0.06 | \+0.24 | \-0.54 | \+0.43 | \-0.04 |
>
>
> 6. __Additional Computational Cost for HOI Cues:__ As shown in the table below, the added branches for HOI cues incur minimal overhead compared to the main vision backbone. We also report the cost of extracting each HOI cue.
>
> __Table: Cost of Additional Layers__
> |  | Extra Params | Extra GFLOPs / video |
> | :---- | :---- | :---- |
> | Hand Pose Branch | 0.80M | 0.0052 |
> | Object BBox Branch | 0.92M | 0.0351 |
> | Object Feat Branch | 1.02M | 0.0602 |
>
>
> __Table: Cost of Feature Extraction__
> |  | Params | GFLOPs / frame |
> | :---- | :---- | :---- |
> | Hand Pose (Hand Detection by YOLOv8 \+ Pose Prediction by WiLoR) | 26.63M \+ 693.03M | 41.56 \+ 140.09 |
> | Object BBox (Faster R-CNN w/ res101) | 47.36M | 219.57 |
> | Object Feats (CLIP) | 202.05M | 51.90 |

---

> > ### Comment · Reviewer_zo7k · 2025-08-07
> >
> > Although initial concerns were raised regarding task formulation, annotation consistency, and evaluation protocol, the rebuttal has sufficiently addressed these issues through clarifications and additional analysis. Given the responses to my initial concerns, I have adjusted my final recommendation to borderline accept.

---

> > > ### Author Response · Authors · 2025-08-07
> > >
> > > Thank you very much for taking the time to thoroughly review our paper and evaluate our rebuttal. We sincerely appreciate your thoughtful feedback and are grateful for your updated recommendation.

---

### Official Review · Reviewer_g84B · 2025-07-04

**Rating:** 4
**Confidence:** 3

**Summary:**

This paper introduces HanDyVQA, a new benchmark for Hand-Object Interaction (HOI), aiming to address the limitations of current HOI benchmarks that focus primarily on high-level action recognition and hand/object localization. HanDyVQA emphasizes the fine-grained spatiotemporal dynamics involved in hand-object interactions, incorporating nuanced details such as hand-object movements, manipulation processes, and state changes. The benchmark includes six types of questions—Action, Process, Objects, Location, State Change, and Object Parts—along with 11.7k multiple-choice question-answer pairs and 11k instance segmentations. These require models to interpret detailed action contexts and interaction dynamics. The paper evaluates various video foundation models, highlighting new challenges in geometric and semantic understanding of fine-grained HOI components.

**Additional Feedback:**

N/A

**Dataset Code Accessibility:**

Partly

**Dataset Code Comments:**

N/A

**Ethical Considerations:**

No, there are no or only very minor ethics concerns

**Final Justification:**

The authors have addressed most of my concerns. As a result. I've decided to keep my initial rating.

**Limitations Weaknesses:**

1. The annotation process appears to be heavily dependent on human input, which could incur significant costs.

2. A few questions remain:

 + Can the proposed dataset be split into training and testing sets?

 + Will the authors release the dataset as open-source?

**Strengths Contributions:**

1. The dataset is valuable for studying hand-object interaction, providing a rich resource for this domain.

2. The experiments quantitatively assess the current model's performance in hand-object understanding.

3. Qualitative results demonstrate the current MLLM's capabilities on this task.

---

> ### Author Rebuttal · Authors · 2025-07-31
>
> We appreciate the reviewer’s recognition of the richness and value of our dataset for advancing the study of hand-object interaction. We are also pleased that the reviewer acknowledged the importance of both our quantitative evaluations and qualitative analyses in assessing the capabilities of current MLLMs using the proposed benchmark.
>
> We answer the reviewer's questions as follows:
>
> 1. __Human annotation costs__: We reduce the overall annotation costs by incorporating large language models (LLMs) in the initial stage instead of relying solely on human annotation. Specifically, LLMs are used to generate a seed set of VQA data, which is then reviewed and refined by human annotators (Sec 3.1). This collaborative approach allowed us to efficiently construct the dataset with a reasonable annotation cost (3,180,000 JPY, approximately 21,400 USD), achieving a balance between scalability and quality.
> 2. __Can the proposed dataset be split into training/testing sets?__: Yes. We provide official training, validation, and test splits in a 10:5:85 ratio (Sec 3.2). All of our experiments followed this split.
> 3. __Will the authors release the dataset as open-source?__: Yes. We have already made our dataset publicly available on Hugging Face. The repository includes all VQA data, including segmentation ground-truth masks, except for the original video data, which can be downloaded from the official Ego4D repository.

---

> > ### Comment · Reviewer_g84B · 2025-08-05
> >
> > Thank you for the authors' response. Since the initial annotation process requires the use of LLMs, I’m curious that how well do current MLLMs/LLMs perform on the proposed task?

---

> > > ### Author Response · Authors · 2025-08-05
> > > **Current MLLMs/LLMs Show Limited Performance on Our Benchmark**
> > >
> > > Thank you for the reviewer’s response.
> > >
> > > Current MLLMs still struggle with our proposed benchmark—even the best-performing model achieves only around 70% accuracy—highlighting the need for improvement in understanding fine-grained hand–object interaction dynamics. Additionally, LLMs alone perform poorly on the task without access to visual information, achieving only around 40% accuracy. Please refer to Section 4.1 for detailed experimental results and ablation studies.
> > >
> > > During the annotation process, LLMs can effectively assist humans in generating distractors, as they are provided with accurate question–answer pairs that have already been verified by human annotators. Since this step does not require visual understanding, LLMs can generate plausible distractors without relying on visual information (which are later refined by human annotators using visual context to make them sufficiently challenging).
> > >
> > > If you have any further questions, please feel free to ask.

---

### Decision · Program_Chairs · 2025-09-18

**Decision:**

Reject

**Comment:**

The paper presents HanDyVQA, a video question-answering benchmark designed to capture the fine-grained spatiotemporal dynamics of hand-object interactions. All reviewers rated the paper as accept, recognizing the dataset’s value, its potential impact on the field, and the comprehensive experimental validation. However, some concerns were raised, including limited question diversity and interaction modeling, ambiguity in task categorization, the absence of failure case analysis, and the cost of human annotation, among other annotation-related issues. The rebuttal effectively addresses most of these major concerns. The Area Chair concurs with the reviewers' consensus and recommends acceptance of the paper.

===== FINAL UPDATE FROM DB Track PCs ====

The final decision for this paper has been taken by the program chairs after consultation with the SACs. All Senior Area Chairs have ranked papers according to the feedback from the AC during the review process. We decided to leave the original meta-review to reflect the opinion of the AC in light of the initial discussions with reviewers and SAC.